# Two-dimensional heavy fermion in a monoatomic-layer Kondo lattice YbCu$_2$

Takuto Nakamura [1,2] ✉, Hiroki Sugihara[2], Yitong Chen [2], Ryu Yukawa [3], Yoshiyuki Ohtsubo [4], Kiyohisa Tanaka [5], Miho Kitamura[6], Hiroshi Kumigashira [7] & Shin-ichi Kimura [1,2,5] ✉

The Kondo effect between localized $f$-electrons and conductive carriers leads to exotic physical phenomena. Among them, heavy-fermion (HF) systems, in which massive effective carriers appear due to the Kondo effect, have fascinated many researchers. Dimensionality is also an important characteristic of the HF system, especially because it is strongly related to quantum criticality. However, the realization of the perfect two-dimensional (2D) HF materials is still a challenging topic. Here, we report the surface electronic structure of the monoatomic-layer Kondo lattice YbCu$_2$ on a Cu(111) surface observed by synchrotron-based angle-resolved photoemission spectroscopy. The 2D conducting band and the Yb $4f$ state, located very close to the Fermi level, are observed. These bands are hybridized at low-temperature, forming the 2D HF state, with an evaluated coherence temperature of about 30 K. The effective mass of the 2D state is enhanced by a factor of 100 by the development of the HF state. Furthermore, clear evidence of the hybridization gap formation in the temperature dependence of the Kondo-resonance peak has been observed below the coherence temperature. Our study provides a new candidate as an ideal 2D HF material for understanding the Kondo effect at low dimensions.

Heavy fermion (HF) systems in rare-earth (RE) intermetallic compounds originating from hybridization between localized $f$-electrons and conduction electrons, namely $c$-$f$ hybridization, are central topics in the field of the strongly-correlated electron systems[1]. At low temperatures, depending on the strength of the $c$-$f$ hybridization, the physical properties change from itinerant $f$ electrons because of the Kondo effect to a magnetic order originating with magnetic moments of localized $f$ electrons due to Ruderman–Kittel–Kasuya–Yosida (RKKY) interactions. The competition between itinerant and localized characters of the $f$-electrons make a quantum critical point (QCP), resulting in the emergence of fertile quantum phenomena such as non-Fermi liquid behavior, and non-BCS HF superconductivity[2,3].

On the other hand, the dimensionality in the system characterizes the fundamental physical property. In low-dimensional systems, the enhancement of the electron-electron correlation and/or breaking of the inversion symmetry leads to novel quantum states such as Rashba-type spin-splitting[4], Tomonaga–Luttinger liquid[5,6], and unconventional superconductivity[7,8]. The combination of the HF state and low dimensionality modifies the ground state of the system because the order parameter of these systems is much more sensitive to dimensionality[9,10]. The ground state of two-dimensional (2D) HF can be easily controlled to the vicinity of a quantum critical point, which is the host to realize unconventional physical properties such as HF superconductivity, by simple external fields such as gate-tuning[11,12], and surface doping[13] in addition to traditional external perturbations;

[1]Graduate School of Frontier Biosciences, Osaka University, Suita 565-0871, Japan. [2]Department of Physics, Graduate School of Science, Osaka University, Toyonaka 560-0043, Japan. [3]Graduate School of Engineering, Osaka University, Suita 565-0871, Japan. [4]National Institutes for Quantum Science and Technology, Sendai 980-8579, Japan. [5]Institute for Molecular Science, Okazaki 444-8585, Japan. [6]Photon Factory, Institute of Materials Structure Science, High Energy Accelerator Research Organization (KEK), 1-1 Oho, Tsukuba 305-0801, Japan. [7]Institute of Multidisciplinary Research for Advanced Materials (IMRAM), Tohoku University, Sendai 980-8577, Japan. ✉e-mail: nakamura.takuto.fbs@osaka-u.ac.jp; kimura.shin-ichi.fbs@osaka-u.ac.jp

temperature, pressure, and magnetic field. Fabricating artificial low-dimensional strongly correlated electron systems and quantizing a three-dimensional HF state by quantum confinement[14] are suitable methods to investigate the novel electronic phase. In the Ce-based artificial superlattice, the suppression of antiferromagnetic (AFM) ordering as well as the increase of the effective electron mass with decreasing of the thickness of the Ce-layer[15] and the emergence of the strong-coupling superconductivity[16] have been reported. To understand the fundamental properties of 2D HF systems, it is necessary to clarify the electronic band structure and the formation mechanism of the HF. However, the details have remained unclear due to the lack of promising materials and the extremely low transition temperatures of less than a few kelvin to HF even in known materials[14,15,17,18].

The growth of the well-ordered atomically thin film on a single crystal substrate is a suitable technique to access such 2D electron systems. So far, various 2D Kondo-lattice has been fabricated on the substrates; multilayer $CePt_5$ thin-film on Pt(111)[19], $CePb_3$ on Si(111)[20], Graphene on $SmB_6$[21], and a checkerboard pattern of organic molecules on Au(111)[22]. However, Yb-based 2D HF material, in which the Yb ion is the most fundamental element to realize HF[23,24] and has a symmetrical electronic-hole configuration to the Ce one, has not been reported. In particular, the RE-based monoatomic layer Kondo-lattice showing HF state has never been reported.

In this study, we report the HF electronic structure of a novel Yb-based monoatomic layer Kondo lattice; synchrotron-based angle-resolved photoemission spectroscopy (ARPES) on monoatomic layered $YbCu_2$ on Cu(111). The surface atomic structure of $YbCu_2$ on Cu(111) is shown in Fig. 1a. The Yb atoms surrounded by Cu atoms are arranged in a triangular lattice. In a similar surface alloy RE $NM_2$/NM(111) (NM = noble metal), various physical properties appear such as ferromagnetic ordering[25–27] and Weyl nodal-line fermion[28] depending on the containing RE element, but there is no report on the appearance of HF character so far. Figure 1c, d shows the LEED patterns

of the Cu(111) substrate and the Yb-adsorbed Cu(111) surface at 70 K, respectively. In addition to the primitive (1 × 1) spots originating from the Cu(111) substrate indicated by yellow arrows, the $(\sqrt{3} \times \sqrt{3})$R30° structure with the Moiré patterns, originating from the small lattice mismatch between Cu(111) and the topmost surface alloy layer, was observed, indicating the successful fabrication of the monoatomic $YbCu_2$ layer, one possible model of the Yb-Cu surface alloy system, on the Cu(111) substrate. Note that the overall trend of the LEED patterns is consistent with those of other RE $NM_2$/NM(111) systems[25–30].

## Results and discussion

The itinerant or localized character of Yb 4$f$ electrons is strongly reflected in the valency of the Yb ions. Figure 2a shows Yb 3$d$ core-level spectrum of $YbCu_2$/Cu(111) at 15 K. The photoelectron peaks at the binding energies of 1528 and 1538 eV originate from the $Yb^{2+}$ and $Yb^{3+}$3$d$ final states, respectively, after photoexcitation. From the intensity ratio between the $Yb^{2+}$ and $Yb^{3+}$ peaks after subtracting the background indicated by the dotted line in the figure, the mean valence of Yb ions was evaluated as 2.41 ± 0.01. To confirm the consistency of the coexistence of $Yb^{2+}$ and $Yb^{3+}$ observed in the Yb 3$d$ core-level spectra to the electronic state near the Fermi level ($E_F$), angle-integrated valence-band photoelectron spectra of the Cu(111) clean substrate and the $YbCu_2$/Cu(111) surface are shown in Fig. 2b. The Cu 3$d$ states at the binding energy of 3 eV are dominant in the Cu(111) substrate. In the $YbCu_2$/Cu(111) spectrum, there are two narrow peaks originating from the $Yb^{2+}$4$f$ spin-orbit pair near $E_F$, and broad peaks of $Yb^{3+}$4$f$ final states and Cu 3$d$ states at the binding energy of 3–13 eV. These results strongly suggest that the Yb ions in monoatomic layer $YbCu_2$ are mixed valence. Note that in the $YbAu_2$/Au(111), which has a similar atomic structure to $YbCu_2$/Cu(111), Yb ions are almost divalent[29]. The reason for the difference in the Yb valence between $YbCu_2$ and $YbAu_2$ would be due to the in-plane lattice compression, which can be explained by the analogy from the bulk Yb-based

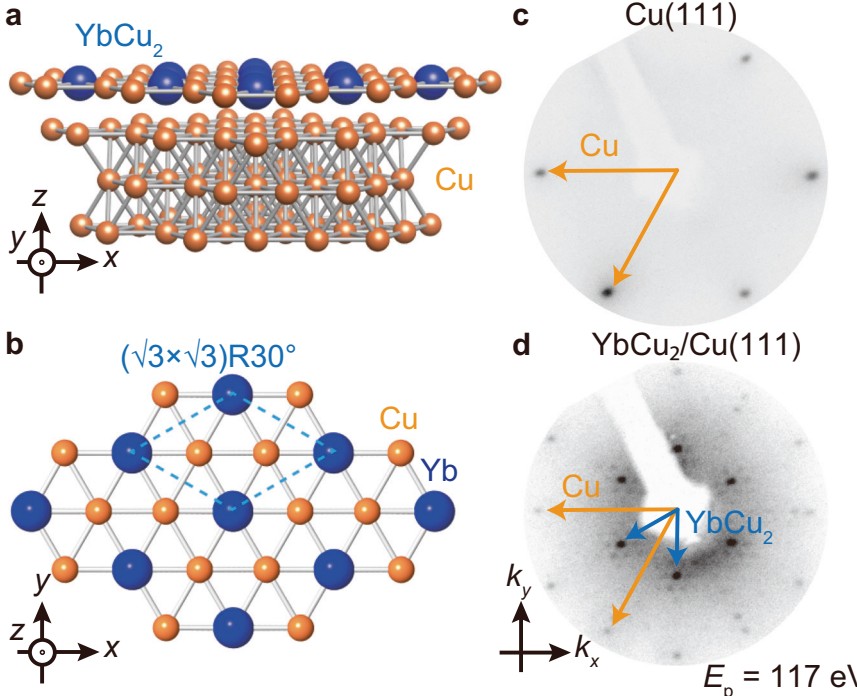

**Fig. 1 | Monoatomic-layer $YbCu_2$ on Cu(111) substrate. a** A surface atomic structure of $YbCu_2$/Cu(111). **b** Top view of monoatomic-layer $YbCu_2$. The dashed line indicates the unit cell of $YbCu_2$. **c** LEED pattern of Cu(111)-(1×1) substrate. **d** Same as (**b**) but for $YbCu_2$/Cu(111)-($\sqrt{3} \times \sqrt{3}$)R30°. Both LEED patterns were taken at the temperature of 70 K. The primitive (1 × 1) and ($\sqrt{3} \times \sqrt{3}$)R30° are indicated by orange and blue arrows, respectively. The distortions of the LEED image are due to the flat microchannel plate used for the LEED measurement. The satellite spots around the integer spots represent the moiré superstructure originating from a small lattice mismatch of $YbCu_2$ and Cu(111).

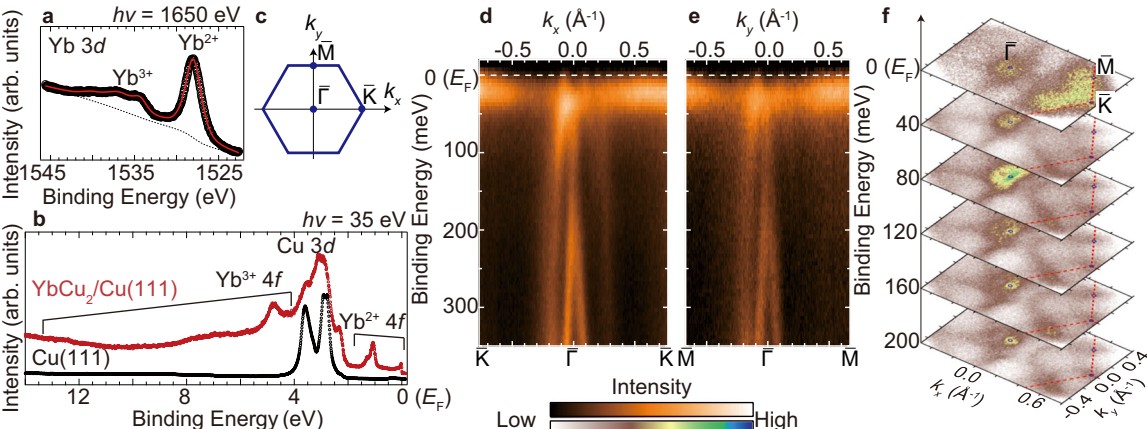

**Fig. 2 | Surface electronic structures of YbCu₂/Cu(111). a** Yb 3*d* core-level spectrum of YbCu₂/Cu(111) taken with 1650-eV photons at the temperature of 15 K. Black circles and red lines represent the raw data and fitted curve, respectively. The dotted line indicates the Shirley-type background. **b** Angle-integrated valence-band spectra of Cu(111) (black) and YbCu₂/Cu(111) (red) taken with horizontally polarized 35-eV photons at 10 K. **c** A hexagonal surface Brillouin zone of YbCu₂/Cu(111). $k_x$ and

$k_y$ are defined along $\bar{\Gamma}$-$\bar{K}$ and $\bar{\Gamma}$-$\bar{M}$ of YbCu₂. **d, e** ARPES intensity plots along $\bar{\Gamma}$-$\bar{K}$ and $\bar{\Gamma}$-$\bar{M}$ taken with horizontally polarized 37-eV photons at 7 K. ARPES intensities are divided by the Fermi−Dirac distribution function convolved with the instrumental resolution. **f** Constant energy contours taken with the energy window of ±15 meV using 35-eV photons at 7 K. The contour at the binding energy of 0 eV corresponds to an experimental Fermi surface.

intermetallic compounds under high pressure[31], because the lattice constant of bulk Cu is about 10% smaller than that of Au. The lattice compression would promote the valence transition from Yb²⁺ to Yb³⁺ due to the smaller ionic radius of Yb³⁺ than that of Yb²⁺, realizing a mixed-valence state in YbCu₂/Cu(111).

Figure 2d, e shows ARPES band dispersions at 10 K along $\bar{\Gamma}$−$\bar{K}$ and $\bar{\Gamma}$−$\bar{M}$, respectively, in the hexagonal surface Brillouin zone (SBZ) shown in Fig. 2c. The flat band is close to $E_F$ and highly dispersive bands are observed near the $\bar{\Gamma}$ point. According to the previous study and the DFT calculation for other RE NM₂/NM(111) families, the flat band and well-dispersive bands mainly originate from the Yb²⁺ 4$f_{7/2}$ and the mixing of the Yb 5*d* and Cu *sp* and *d* orbitals, respectively[26,29,32]. The detailed assignments of these bands are shown in Supplementary Note 1. It should be noted that the photoelectron intensities of the dispersive band near the $\bar{\Gamma}$ point at the positive wavenumber region are relatively weak in both $\bar{\Gamma}$−$\bar{K}$ and $\bar{\Gamma}$−$\bar{M}$ directions due to a photoexcitation selection rule[33]. The energy position of the Yb²⁺ 4$f_{7/2}$ is very close to $E_F$, which is the general feature of the Yb-based HF system such as YbRh₂Si₂[23] and β-YbAlB₄[24], suggesting the same mixed valent character of Yb ions in YbCu₂/Cu(111) as the result of Yb 3*d* peaks. It should be noted that the surface electronic structure of almost all HF materials tends to be localized, which is inconsistent with that of the bulk, due to a surface lattice expansion. Also, the Yb-ions in bulk YbCu₂ are mixed-valence[34,35], which is similar to YbCu₂/Cu(111), but the orthorhombic crystal structure is different from that of YbCu₂/Cu(111). Additionally, the surface state of bulk YbCu₂ is divalent, which is consistent with other Yb compounds. Therefore, the origin of the mixed-valent character of YbCu₂/Cu(111) is not the same as that of bulk YbCu₂.

In Figure 2d, e, the Yb²⁺ 4*f* flat band is modulated at the cross points to the conduction bands just below $E_F$, red providing evidence of *c-f* hybridization[36]. It should be noted that the *c-f* hybridization bands can appear in periodically located Yb and Cu atoms on the surface, not in randomly diluted Yb impurities in bulk Cu. The hole bands at the $\bar{\Gamma}$ point and the Yb²⁺ 4$f_{7/2}$ states near $E_F$ can be confirmed to originate from the YbCu₂ layer by calculations[26,29,32] (see Supplementary Fig. S2). Figure 2f shows a series of the constant energy contours of the YbCu₂/Cu(111) surface. In the map at the binding energy of 0 eV, which corresponds to the experimental Fermi surfaces, there are strong photoelectron intensity areas at $\bar{\Gamma}$ point and near the zone boundary. However, as shown in Fig. 2d, e, non-bonding 4*f*

character only appears near the $\bar{K}$ and $\bar{M}$ points, but the *c-f* hybridization feature only exhibits near the $\bar{\Gamma}$ point. Therefore, in the following part, we focus on the hybridized band near the $\bar{\Gamma}$ point to investigate in detail the HF character appearing in monoatomic layered YbCu₂.

In the HF system, the size of the Fermi surface is modulated by the changing of the temperature due to the enhancement of the *c-f* hybridization. Figure 3a shows the temperature-dependent ARPES images along $\bar{\Gamma}$−$\bar{K}$. The overall trend of those images is consistent except near $E_F$. To reveal the change of the dispersion at $E_F$ in more detail, the momentum-distribution curves (MDCs) at $E_F$ are plotted in Fig. 3b. The peak position of the hybridized band near the $\bar{\Gamma}$ point at 130 K is shifted toward a higher wavenumber at 7 K, suggesting an enlargement of the Fermi surface by the development of the *c-f* hybridization at low temperature.

Figure 3c shows the ARPES image around the $\bar{\Gamma}$ point taken with circularly polarized photons at 15 K. The *c-f* hybridization branches S1 and S2 are visible. To determine the dimensionality of the *c-f* hybridization bands, the photon-energy dependence of ARPES was measured as shown in Fig. 3d. Both S1 and S2 bands show no photon-energy dependence, indicating no out-of-plane ($k_z$) dispersion. These experimental results strongly suggest that the *c-f* hybridization band is formed in the 2D YbCu₂ plane. To evaluate the *c-f* hybridization feature, comparing it to the periodic Anderson model (PAM) is helpful[37]. In the case of the Coulomb repulsion energy between 4*f* electrons $U_{ff}$ is zero or infinity, the band dispersions $E_k^{\pm}$ of PAM is described as

$$E_k^{\pm} = \frac{\epsilon_c + \epsilon_f \pm \sqrt{(\epsilon_c - \epsilon_f)^2 - 4V_k^2}}{2} \quad (1)$$

where $\epsilon_c$ and $\epsilon_f$ are the dispersions of the conduction band and the 4*f* band, respectively, and $V_k$ is the hybridization intensity. For the fitting using the PAM, a *k*-linear hole band dispersion was assumed for the conduction band to reproduce the steep band shape observed by ARPES. From the fitting by Equation (1), $\epsilon_f$ and $V_k$ are evaluated as 0.06 and 0.12 eV, respectively. The fitting results are shown in Fig. 3c. The solid and dashed lines indicate the band dispersions $E_k^{\pm}$ with $V_k = 0.12$ eV and 0 eV, respectively. From the PAM analysis, the Fermi velocity $v_F$ and Fermi wavenumber $k_F$ of the bare unhybridized conduction band are evaluated as 4.77 eV Å and 0.004 Å⁻¹, respectively. The effective mass of bare conduction band $m_b$ becomes $5.65 \times 10^{-33}$ kg. Note that the shape of the simulated bare conduction band is good agreement

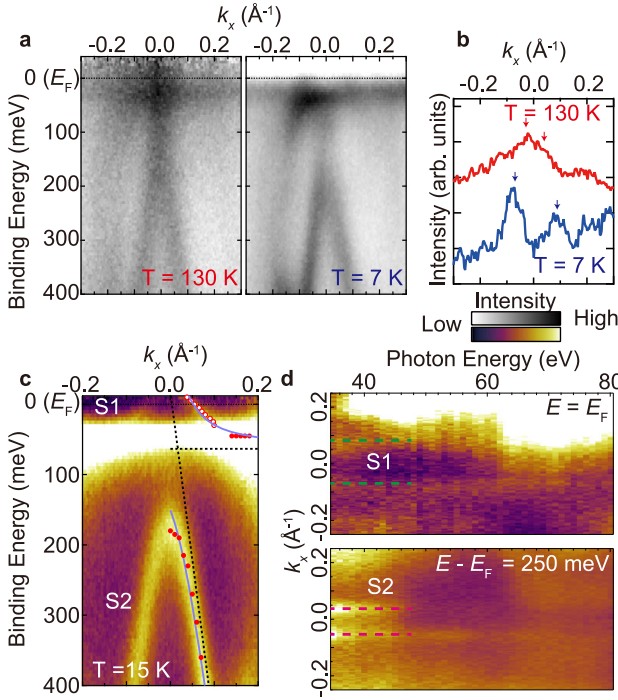

**Fig. 3 | Two-dimensional heavy fermion state in YbCu$_2$/Cu(111). a** ARPES intensity plots along $\bar{\Gamma}$–$\bar{K}$ at 130 and 7 K taken with horizontally polarized 37-eV photons. ARPES intensities are divided by the Fermi–Dirac distribution function convolved with the instrumental resolution. **b** Momentum distribution curves at $E_F$ taken from (**a**) with the energy windows of ±10 meV. Arrows indicate the peak positions of the MDCs. **c** Magnified ARPES image near the $\bar{\Gamma}$ point taken with circularly polarized 35-eV photons at 15 K. ARPES intensities are divided by the Fermi–Dirac distribution function convolved with the instrumental resolution. The filled and break lines indicate the simulated band dispersions $E_k^{\pm}$ with $V_k$ = 120 meV and 0 meV by the PAM. The open and filled circles indicate the peak positions from energy distribution curves (EDCs) and MDCs, respectively. **d** Photon-energy dependence of MDCs at the normal emission at binding energies of 0 eV (upper panel) and 250 meV (lower panel) with the energy windows of ±10 meV. Dashed lines indicate the guide of the MDC peak position by eye.

with the ARPES image at room temperature as shown in Fig. S7 in Supplementary Note 5. On the other hand, by the fitting of the ARPES band dispersion near $E_F$ at 15 K, $v_F$ and $k_F$ of the 2D HF band are 0.58 eV Å and 0.055 Å$^{-1}$, respectively, and the effective mass $m^*$ of the HF band is $6.56 \times 10^{-31}$ kg. By comparing the experimentally obtained heavy conduction band at 15 K and the simulated bare conduction band, a mass enhancement factor of the effective mass $m^*$ of the HF state at 15 K from the unhybridized one $m_b$ ($m^*/m_b$) is evaluated as about 120 suggesting the appearance of heavy quasiparticles at low temperatures.

We now discuss the temperature dependence of the quasiparticle peak just below $E_F$, so-called Kondo resonance (KR) peak. In the HF system, the temperature dependences of the energy position and intensity of the KR peak are reflected in the spectral weight transfer between 4$f$ state and the conduction band as well as renormalization due to the development of the $c$-$f$ hybridization. Figure 4a1 shows the angle-integrated photoelectron intensity near $E_F$ as a function of temperature, and Fig. 4a2 is the same, but the intensities are divided by the Fermi–Dirac distribution function convolved with the instrumental resolution. The KR peak energy is shifted to the $E_F$ side with decreasing temperature, indicating the evolution of the renormalization due to the HF formation. To discuss the temperature-dependent development of the HF state in more detail, the peak positions and intensities are obtained from the fitting of Fig. 4a2 by a Lorentzian function after subtracting a Shirley-type background, as shown in Fig. 4d, e. The peak

position shifts from 42 meV to 22 meV with decreasing temperature and is saturated at 30 K. The integrated intensity increases with decreasing temperature and is also saturated at 30 K. According to photoelectron spectroscopic studies of bulk RE intermetallic compounds, such saturated temperature represents a coherence temperature ($T_{coh}$), at which the $c$-$f$ hybridization state is fully established, resulting in a HF state[38,39]. $T_{coh}$ sets the temperature scale on which the weight increases and saturation typically occurs below $T_{coh}$. This result is evidently visible in spectral properties like ARPES, while response quantities, such as conductivity and specific heat, are less clear. It has been shown by Bethe ansatz solution of the single-impurity problem and numerous calculations for lattice problems[40]. The spectral weight is strongly suppressed towards $E_F$, even after the spectra are divided by the Fermi–Dirac distribution function, suggesting that the $c$-$f$ hybridization gap appears below $T_{coh}$. The above experimental fact strongly suggests that YbCu$_2$ is the 2D monoatomic-layered HF material with $T_{coh} \sim 30$ K, which is more valence fluctuative than other low-dimensional HF such as CeIn$_3$/LaIn$_3$ superlattice ($T_{coh}$ = 1.6 K)[15].

To investigate the momentum-dependent $c$-$f$ hybridization formation, we took the temperature-dependent peak position and intensity of the quasiparticle peak at three wavenumbers ($k_x$ = 0.5, 0.0, −0.1 Å$^{-1}$) as shown in Fig. 4d, e. The change of the peak position at $k_x$ = 0.5 Å$^{-1}$, which is the cut only the Yb$^{2+}$ 4$f_{7/2}$ state, almost follows the angle-integrated one. This suggests that the angle-integrated spectrum mainly focuses on the high-density Yb$^{2+}$ 4$f$ state and the renormalization is effective for all 4$f$ states. In contrast to the saturated feature in the angle-integrated spectrum at $T$ = 30 K, the KR peak positions at $k_x$ = 0.0 and −0.1 Å$^{-1}$ are shifted toward the higher-binding energy side below $T_{coh}$ suggesting the hybridization gap enlargement, even though the peak shifts at $k_x$ = 0.0 and −0.1 Å$^{-1}$ also follows the angle-integrated peak above $T_{coh}$. On the other hand, the peak intensity, which corresponds to the spectral weight transfer between Yb 4$f$ states and conduction bands, continued to increase below $T_{coh}$ as shown in Fig. 4e. To the best of our knowledge, similar behavior has not been reported yet except for the angle-integrated photoemission spectroscopy of a Kondo semiconductor SmB$_6$[41,42], which originated from the transition from the metallic to the semiconducting state, but the magnitude of the peak shift observed here is much larger than that of SmB$_6$. The peak shift in SmB$_6$ is immediately saturated just below $T_{coh}$, which is not consistent with the behavior in YbCu$_2$, suggesting a different mechanism of the KR peak shift. Both the peak shift and the developing intensity below $T_{coh}$ of YbCu$_2$ suggest that the 2D HF state still develops even below $T_{coh}$. Further theoretical analysis such as the dynamical mean field theory about the development of the HF state in the 2D system would help the understanding of the temperature-dependent behavior of the KR peak.

In conclusion, we report the electronic structures of a novel Yb-based monoatomic layered Kondo lattice YbCu$_2$ on Cu(111) by ARPES. Our spectroscopic data provide direct evidence of the appearance of a purely 2D HF state with $T_{coh}$ = 30 K, which is extremely higher than other 2D HF materials[15,17,18], in a monoatomic layered material for the first time. Monoatomic layered YbCu$_2$ is the minimal material to realize low-dimensional HF containing RE elements and act as a building block to reveal novel electron-correlation-driven phenomena, for example, the proximity effect of layered material between 2D HFs and other many-body interactions such as superconductivity and magnetism. Quantum fluctuations in 2D materials are much more sensitive to external fields[18]. The ground state of YbCu$_2$ would be tuned around QCP by external perturbations conventionally applied to other 2D materials, where surface carrier doping by alkali metal adsorption and gate-tuning of carrier concentration by the growth of Cu(111) ultrathin film on an insulating substrate such as sapphire may be realized. Such novel techniques are expected to explore novel quantum critical phenomena in 2D materials, such as atomic-layer unconventional superconductivity.

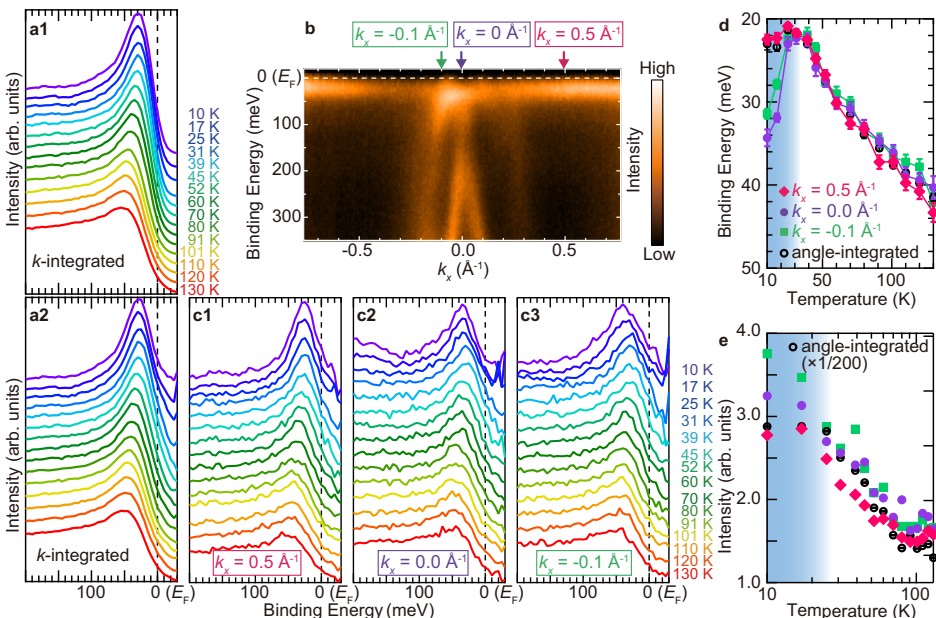

**Fig. 4 | Momentum dependence of temperature-evolution of Kondo resonance peak by temperature. a1** Angle-integrated photoelectron spectra near $E_F$ as a function of temperature taken with horizontally polarized 35-eV photons. **a2** same as (**a1**) but the intensity is normalized by the Fermi−Dirac distribution function convolved with the instrumental resolution. **b** The same ARPES image as Fig. 2d indicates the $k_x$ positions where the temperature-dependent peak energy and intensity were measured. **c** Angle-resolved photoelectron spectra near $E_F$ as a function of the temperature. The $k_x$ positions at 0.5, 0.0, and −0.1 Å⁻¹ are representative of the local $f$ state only, the $\bar{\Gamma}$ point, and the crossing point of the $c$-$f$ hybridization, respectively. **d** Momentum dependence of the energy position of the quasiparticle peak plotted on a linear scale of temperature. **e** Momentum dependence of the intensity of the quasiparticle peak plotted on a logarithmic scale of temperature. The shaded area indicates the HF state estimated by the saturated temperature in angle-integrated data, and the boundary between the shaded area and the white background region indicates the experimentally evaluated coherence temperature $T_{coh}$. Error bars are determined by the standard deviation of the fits to the data.

## Methods

### Sample preparation
Cu(111) substrate was cleaned by Ar-ion sputtering with an acceleration energy of 0.5 keV and annealing at 800 K. After several sputtering and annealing cycles, a sharp (1 × 1) low-energy electron diffraction (LEED) pattern was confirmed as shown in Fig. 1c. Yb atoms were evaporated on the Cu(111) substrate at 600 K. Because the crystallinity of the YbCu$_2$ layer was quite sensitive to the substrate condition such as cleanness of surface and growth temperature, we precisely monitored the sharpness of the diffraction from YbCu$_2$ and moiré pattern, which is directly linked to the crystal quality of the YbCu$_2$/Cu(111), by the reflective high energy electron diffraction (RHEED) in the growth process.

### Photoemission experiments
ARPES and core-level photoemission spectroscopy measurements were performed at BL-2A MUSASHI of the Photon Factory, and BL7U SAMRAI[43] of the UVSOR-III Synchrotron Facility. The energy resolution and the energy position of the Fermi-level were calibrated by the Fermi-edge of polycrystalline Au films electrically contacted to the sample holder. Energy resolutions for ARPES and core-level photoemission spectroscopy were better than 20 meV and 100 meV, respectively. In temperature-dependent measurements, the position of $E_F$ and the instrumental resolution were accurately calibrated by measuring the Fermi edge of the Au thin film at all measurement temperatures.

### Band calculations
Band structures of freestanding YbCu$_2$ and YbCu$_2$/Cu(111) slab were calculated by using the WIEN2K code[44] including spin-orbit interaction within the generalized gradient approximation of the Perdew, Burke, and Ernzerhof exchange-correlation potential[45]. The in-plane lattice

constant of the YbCu$_2$ was set to the experimentally obtained value (4.80 Å) from LEED measurements. The atomic structure of YbCu$_2$/Cu(111) was modeled by a symmetric slab of six layers of Cu with a surface covered with YbCu$_2$ layers. No electron correlation was included, in the band calculations. To obtain the overall trend of the electronic structure, such a condition that does not include the electron correlation would also be sufficient. The calculated band structures are shown in Figs. S1 and S2.

## Data availability
The datasets generated during and/or analyzed during the current study are available from the corresponding author upon request.

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

## Acknowledgements

We acknowledge M.F. Lubis and K. Nishihara for their technical support during the experiments. We would like to thank Professors Takahiro Ito and Hiroshi Watanabe for the helpful discussions. The ARPES measurements were partially performed under UVSOR proposals 22IMS6861, 22IMS6848, and Photon Factory proposal 2022G513. This work is supported by JSPS KAKENHI (Grants Nos.22K14605 (T.N.), 20H04453 (S.K.), and 23H00090 (S.K.)).

## Author contributions

T.N., H.S., and Y.C. conducted the ARPES experiments with assistance from R.Y., K.T. M.K., and H.K.; T.N and Y.O. performed the DFT calculations; T.N and S.-i.K. wrote the text and were responsible for the overall direction of the research project. All authors contributed to the scientific planning and discussions.

## Competing interests

The authors declare no competing interests.
