## [Peer Review File · Nature Communications]

Reviewers' Comments:

Reviewer #1:

Remarks to the Author:

Manuscript Number : NCOMMS-23-27574-T

Title of Article :

~~Two-dimensional~~ heavy fermion in a monoatomic-layer Kondo lattice YbCu₂"

Authors: Dr Nakamura and colleagues

Comments and issues.

The authors have prepared a monoatomic layer of YbCu₂ on Cu(111) substrate and performed spectroscopic measurements by angle resolved photoelectron spectroscopy (ARPES). The main claim of the authors is that this is the first of its kind in Yb-based compounds where they confirm the strong hybridization of the conduction electron and the f-electron, thus leading to a heavy fermion state from the ARPES study of a monoatomic layer YbCu₂. The mixed valence state in YbCu₂/Cu(111) is confirmed by the intensity ratio of the Yb²⁺ and Yb³⁺ peaks of the core-level spectrum. From the comparison of the heavy conduction band at 15 K and the simulated bare conduction band the authors claim a mass enhancement factor of about 120, suggesting a heavy quasiparticles in YbCu₂/Cu(111). They have estimated the Kondo coherence temperature as 30 K and the 2D heavy fermion state develops below this temperature. Overall, this study is unique and interesting. The interest in this kind of two dimensional materials is getting attention these days and this is one of its kind on the Yb based compounds.

The article appears sound enough. The results are presented and discussed in a relatively clear manner. However, before the manuscript might be considered for publication the authors have to take into account some remarks and comments as in the following.

I have the following concerns which needs to be addressed.

1. There are several typos in the manuscript. For example, for "arranged" the authors have written "arrayed", similarly for "arrows", they have written "allows".
2. The language of the manuscript warrants a thorough reading and improvement
3. A theoretical comparison of the experimental data will be an added advantage to the manuscript
4. The main conclusion of this study is the HF state in the monoatomic layer, YbCu₂. Hence a more detail on the effective mass calculation would be beneficial in the main text.
5. There are two types of structure for YbCu₂. It crystallizes in the orthorhombic crystal structure with the space group Imma (74) and also in the hexagonal P63/mmc (194). In general, this compound is synthesized in the orthorhombic crystal structure. How did the authors manage to stabilize the monoatomic layers of YbCu₂ in the hexagonal crystal structure.
6. All over the References, all the chemical formulas (*e.g.* : ceni₃, nbse₂, must be properly re-written.

=====

Reviewer #2:

Remarks to the Author:

This work reports the formation of a heavy fermion state in monoatomic YbCu₂ on Cu(111). They characterize the sample using LEED measurements and valence band spectra that show the existence of a mixed valence state between Yb⁺² and Yb⁺³. Synchrotron based ARPES measurements as a function of the temperature provide some evidence for the presence of a flat band at low temperatures, near the Fermi level, and the enhancement of the hybridization between conduction and f bands as temperature decreases, which are signatures of heavy fermion physics. The authors estimate a coherent temperature of 30 K and mass enhancement factor of 100.

The general introduction reads well although some improvements could be made as suggested below. The quality of the data is good and the authors provide supplementary information to support some of the findings discussed in the main text. I consider that the data analysis was carried out correctly, to the best of my knowledge. The results are interesting and may be important for future engineering of 2D heavy fermion states.

I have a number of questions regarding various details of the experiment and analysis. If the authors can properly address these questions, as well as the other remarks listed below, I recommend publication in Nature Communications.

- No information is provided about the properties of bulk YbCu₂ and how these change in the monolayer limit. Yb ions in both, bulk and monolayer YbCu₂, have mixed-valent character but how dimensionality modifies low-energy band hybridization and the Kondo effect is unclear. How does the surface state in bulk YbCu₂ compare with the results obtained in this work?

- It would be useful to directly compare the ARPES data with the band structure calculations provided in the supplementary information. The flat band in the calculations seems to appear at a much larger energy than in the experiments.

- How are the red points in Fig. 3c determined for the upper branch S1 (energies above 100 meV)? The signal intensity appears to be saturated in this energy range. The second derivative of ARPES data is often used to sharpen dispersive data features (see for instance Nature Physics 5, 637–641 (2009)). I think it would be useful to represent the data in this way to better resolve the temperature induced c-f hybridization.

- The value of the effective mass obtained at low temperatures ($m^* = 6.56 \times 10^{-31}$) is smaller than the free electron mass. This makes the use of the term "heavy fermion" difficult to understand and should be explained by the authors. They claim to obtain a mass enhancement factor of about 120. They compare ARPES data at 300 K shown in Fig. S6 and those at 15 K in Fig. 3d, but I can't see much of a difference, particularly in the lower branch S2. Is the effective mass at low temperatures obtained from the upper branch S1? I think the authors should better explain how they have obtained the mass values for the bare and hybridized bands and include these values in the main text.

- I think it would be helpful if the authors included in the supplementary information the ARPES intensity plots used to obtain Fig. 4. This will allow the reader to better assess the evolution of the band hybridization across the coherent temperature.

- The authors state that "Perfect two-dimensional (2D) HF materials have not been reported yet". Although they cited some previous work reporting the fabrication of artificial low dimensional strongly correlated electron systems, other topical references are lacking. For example, recent work published in Nature [Nature 616, 465–469 (2023)] reported lateral quantization of electronic states on the surface of a heavy fermion showing 2D heavy fermion behavior. I think this work should be mentioned to improve the introduction to this topic.

Minor change:

- The sentence "The fitting results are shown in Fig. 2(c)" should be changed to "The fitting results are shown in Fig. 3(c)".

Reviewer #3:

Remarks to the Author:

The authors fabricate a 2D YbCu₂ heavy-fermion lattice on a clean Cu[111] surface by sputtering Yb atoms. LEED measurements confirm the high quality of the surface lattice. The authors then present a comprehensive ARPES study of this system which confirms the heavy-fermion behavior of this system. In particular, they demonstrate the k-dependent dispersion of the heavy band, the temperature dependent change of the Fermi momentum near the Gamma point, and the two-dimensionality of the heavy (flat) part of the band, while the conduction band must obviously be three-dimensionally due to the 3D Cu metal substrate. The measurements are of high quality, and the present system is one of the few truly two-dimensional heavy-fermion systems realized so far.

Therefore, I would in principle recommend publication of this manuscript, however would like to ask for clarification of some points.

(1) Apart from finding the absence of HF dispersion in the z direction (perpendicular to the surface), do the authors have independent evidence that the YbCu₂ lattice is truly a monolayer, as claimed?

(2) Yb is a hole system (13 electrons, i.e., 1 hole, in the 4f shell in the pristine form). The hole-like nature is confirmed by the fact that the (k-integrated) ARPES spectra show the Kondo resonance below the Fermi energy, not above as for particle-like systems, which makes it clearly observable by ARPES, as seen in Fig. 4. Furthermore, the authors observe that the Fermi surface *expands* with lowering the temperature below the (Kondo) lattice-coherence temperature (Fig. 3b). Both observations together imply that also the conduction band is hole-like (this follows from counting mobile charge carriers or from an analysis where hybridized band must cross the Fermi level) and the fitting of the hybridized two-band structure (Eq. (1)) indeed yields a hole-like conduction band. By contrast, Cu is known to have particle-like conduction bands. Do the authors have an explanation for this apparent discrepancy? As this is an obviously emerging question, can they somewhat elaborate on it in the paper?

(3) Fig. 4 a, b shows a strong (almost temperature-independent below 30 K) decrease of spectral weight towards the Fermi energy, even after the spectra are divided by the Fermi-Dirac distribution. This seems to be a clear indication of a hybridization gap or even a complete gap, since this decrease occurs even in the k-integrated spectra. This should be discussed more explicitly in the paper.

(4) The authors claim that gating would be an additional tuning parameter for 2D heavy-fermion systems. For the present system this would be difficult since the substrate is metallic. The authors should weaken their claim at least for the present system.

(5) The authors find an increase of the Kondo spectral weight even at temperatures below the coherence (or Kondo) temperature and quote this as unusual. In fact however, the coherence temperature T_{coh} sets the temperature scale on which the weight increases, and saturation typically occurs below T_{coh} . This is particularly visible in spectral properties like ARPES, while response quantities, like conductivity, specific heat etc, are less clear. It has been shown by Bethe ansatz solution of the single-impurity problem and numerous calculations for lattice problems.

Johann Kroha

Reviewer #1 (Remarks to the Author):

The authors have prepared a monoatomic layer of YbCu₂ on Cu(111) substrate and performed spectroscopic measurements by angle resolved photoelectron spectroscopy (ARPES). The main claim of the authors is that this is the first of its kind in Yb-based compounds where they confirm the strong hybridization of the conduction electron and the f-electron, thus leading to a heavy fermion state from the ARPES study of a monoatomic layer YbCu₂. The mixed valence state in YbCu₂/Cu(111) is confirmed by the intensity ratio of the Yb²⁺ and Yb³⁺ peaks of the core-level spectrum. From the comparison of the heavy conduction band at 15 K and the simulated bare conduction band the authors claim a mass enhancement factor of about 120, suggesting a heavy quasiparticles in YbCu₂/Cu(111). They have estimated the Kondo coherence temperature as 30 K and the 2D heavy fermion state develops below this temperature. Overall, this study is unique and interesting. The interest in this kind of two dimensional materials is getting attention these days and this is one of its kind on the Yb based compounds.

The article appears sound enough. The results are presented and discussed in a relatively clear manner. However, before the manuscript might be considered for publication the authors have to take into account some remarks and comments as in the following. I have the following concerns which needs to be addressed.

Reply: First of all, let us express our deepest acknowledgment of the voluntary efforts of Reviewer #1 for our manuscript. We are encouraged that the reviewer gives a high evaluation of our work as

“Overall, this study is unique and interesting. The interest in this kind of two dimensional materials is getting attention these days and this is one of its kind on the Yb based compounds. The article appears sound enough. The results are presented and discussed in a relatively clear manner.”. We answer the comments, as shown in the following part.

1. There are several typos in the manuscript. For example, for "arranged" the authors have written "arrayed", similarly for "arrows", they have written "allows".

2. The language of the manuscript warrants a thorough reading and improvement

Reply: Thank you for your careful review of our manuscript. We are very sorry for several typos and wrong sentences in the manuscript. We rechecked and revised typos and language as follows:

(main text)

- The Yb atoms surrounded by Cu atoms are **arranged** in a triangular lattice.
- In a similar surface alloy RENM₂/NM(111) (NM = noble metal), various physical properties appear such as **ferromagnetic** ordering and Weyl nodal-line fermion depending on the

containing RE element, but there is no report on the appearance of HF character so far.

- In addition to the primitive (1x1) spots originating from Cu(111) substrate indicated by yellow arrow, the $(\sqrt{3}\times\sqrt{3})R30^\circ$.

(Figure caption)

- The primitive (1x1) and $(\sqrt{3}\times\sqrt{3})R30^\circ$ are indicated by orange and blue arrows, respectively.

(Fig. 1)

- Angle-resolved photoelectron spectra near E_F as a function of the temperature. (Fig. 4, Fig. S4)

3. A theoretical comparison of the experimental data will be an added advantage to the Manuscript

Reply: We agree with Reviewer #1's comments. The comparison between ARPES and DFT is useful for a deeper understanding of the band structure of YbCu_2 . The calculated band structure of $\text{YbCu}_2/\text{Cu}(111)$ compared with the ARPES data taken with horizontally polarized 37-eV photons is shown in Figure R1 with the same scale. Although the calculated band structure is slightly different from the experimental ARPES data, the overall trend of the band structure is qualitatively consistent with the observed ARPES images. So, we could attribute the orbital components of the band dispersions. The innermost hole band is mainly contributed from the YbCu_2 layer, as shown in Fig. S2 (b2), and the broad and weak hole bands located outside of the YbCu_2 band originated from the Cu substrate from Fig. S2 (b3). It is consistent with the $h\nu$ -dependent measurement because the innermost hole band of YbCu_2 shows a 2D character even though the outer hole bands have 3D-like dispersions. We have added the comparison between DFT and ARPES results to Supplementary Note 1.

Figure R1. The comparison of electronic structure between calculation and ARPES. (a)

Calculated band structure of the YbCu₂/Cu(111) slab. (b) ARPES intensity plots along Γ - \bar{K} with horizontally polarized 37-eV photons at 7 K.

4. The main conclusion of this study is the HF state in the monoatomic layer, YbCu₂. Hence a more detail on the effective mass calculation would be beneficial in the main text.

Reply: Thank you for the valuable suggestion. We added the explanation of the evaluation method of the effective mass in the main text from the supplemental note with an additional explanation.

5. There are two types of structure for YbCu₂. It crystallizes in the orthorhombic crystal structure with the space group Imma (74) and also in the hexagonal P63/mmc (194). In general, this compound is synthesized in the orthorhombic crystal structure. How did the authors manage to stabilize the monoatomic layers of YbCu₂ in the hexagonal crystal structure.

Reply: As Reviewer #1 said, the orthorhombic YbCu₂ is usually synthesized as a bulk crystal. In this study, we used the surface alloying technique to make hexagonal YbCu₂ with monoatomically thin thickness. It is well known that the monoatomic layered surface alloy, which shows a hexagonal-like $\sqrt{3}\times\sqrt{3}$ periodicity as shown in Fig. 1(a), appears when a small amount (typically sub-monolayer thickness) of other element deposits on a (111) plane of the atomically clean noble-metal substrate. Then we could make monoatomic layered YbCu₂ with the hexagonal structure by choosing Yb and Cu as the adsorbed element and substrate, respectively.

6. All over the References, all the chemical formulas (e.g. : cen₃, nbse₂, must be properly re-written.

Reply: Thank you for your careful review. We revised all chemical formulas in the Reference section. we could make improvements to our manuscript.

Reviewer #2 (Remarks to the Author):

This work reports the formation of a heavy fermion state in monoatomic YbCu₂ on Cu(111). They characterize the sample using LEED measurements and valence band spectra that show the existence of a mixed valence state between Yb⁺² and Yb⁺³. Synchrotron based ARPES measurements as a function of the temperature provide some evidence for the presence of a flat band at low temperatures, near the Fermi level, and the enhancement of the hybridization between conduction and f bands as temperature decreases, which are signatures of heavy fermion physics. The authors estimate a coherent temperature of 30 K and mass enhancement factor of 100.

The general introduction reads well although some improvements could be made as suggested below. The quality of the data is good and the authors provide supplementary information to support some of the findings discussed in the main text. I consider that the data analysis was carried out correctly, to the best of my knowledge. The results are interesting and may be important for future engineering of 2D heavy fermion states.

I have a number of questions regarding various details of the experiment and analysis. If the authors can properly address these questions, as well as the other remarks listed below, I recommend publication in Nature Communications.

Reply: Let us express our deepest acknowledgment of the voluntary efforts of Reviewer #2 for our manuscript. We are encouraged that the reviewer evaluates our work as "The quality of the data is good and the authors provide supplementary information to support some of the findings discussed in the main text. I consider that the data analysis was carried out correctly, to the best of my knowledge. The results are interesting and may be important for future engineering of 2D heavy fermion states". We answer the questions, as shown in the following part.

- No information is provided about the properties of bulk YbCu₂ and how these change in the monolayer limit. Yb ions in both, bulk and monolayer YbCu₂, have mixed-valent character but how dimensionality modifies low-energy band hybridization and the Kondo effect is unclear. How does the surface state in bulk YbCu₂ compare with the results obtained in this work?

Reply: Thank you for the important comments about comparing bulk and monoatomic layer YbCu₂. The dimension reduction may enhance the electron correlation and magnetic frustration, resulting in the system being close to the quantum critical point due to the suppression of the hybridization. The mean valence of bulk YbCu₂ is ~ 2.2 [A Fujimori PRB 2022], indicating mixed valent character. However, it should be noted that the atomic structure of the bulk YbCu₂ is much different from monolayer YbCu₂, although both bulk and

monoatomic YbCu₂ show a mixed valent character. Therefore, it is not simple to consider continuous connections from the bulk to the monolayer limit in the YbCu₂ system.

On the other hand, the surface state of the orthorhombic bulk YbCu₂ shows a localized character due to the surface lattice expansion generally seen in the surface state of bulk rare-earth intermetallic compounds. The energy position of the surface Yb 4*f* components is a shift to a higher binding energy side [E. J. Cho PRB (1993)]. In other words, the surface state of bulk heavy-fermion materials does not behave as a heavy-fermion character. Therefore, we'd like to emphasize that the mixed valency of monoatomic layered YbCu₂ is completely different from that of the surface on bulk YbCu₂.

- It would be useful to directly compare the ARPES data with the band structure calculations provided in the supplementary information. The flat band in the calculations seems to appear at a much larger energy than in the experiments.

Reply: Thank you for the useful suggestion. Reviewer #1 gave us a similar suggestion. Then, we added the comparison of the ARPES data with the band structure calculations with the same energy and momentum scales in Supplemental Note1. As pointed out by Reviewer #2, the energy position of the Yb 4*f* states of ARPES is shifted to the lower-energy side than that of the DFT calculation. Applying the on-site Coulomb repulsion energy U cannot reproduce the energy position in ARPES because the strong on-site interaction pushes the 4*f* state away from E_F . Also, the relative position between the Yb 4*f*_{5/2} and the flat Cu 3*d* is inverted in Figure S6, i.e., the flat Cu 3*d* state is pushed up to E_F . These imply that the self-energy of the Cu finite contributed to the formation of heavy quasiparticles in monolayer YbCu₂. Further complex analysis, such as calculation of one-particle spectral function and resonance and/or ultra-high-resolution measurement, is considered to be needed to understand such effect.

- How are the red points in Fig. 3c determined for the upper branch S1 (energies above 100 meV)? The signal intensity appears to be saturated in this energy range. The second derivative of ARPES data is often used to sharpen dispersive data features (see for instance Nature Physics 5, 637–641 (2009)). I think it would be useful to represent the data in this way to better resolve the temperature induced c-f hybridization.

Reply: Thank you for the comments. As the reviewer said, the second-derivative plot of the ARPES image is a useful technique for better data representation. Figure R2 shows the ARPES second-derivative plots. The data were derived along (a) energy- and (b) momentum-directions. In Fig. R2(a) (R2(b)) clearly represents the conduction (4*f*) band. In the manuscript, to obtain the energy- and momentum position of the c-*f* hybridization band at the E_F , we extracted the peaks of the raw data of MDCs shown in Fig. R2(c). While certainly

less visibility than the second-derivative plots may be, we could trace the peak positions of all the raw spectra. Figure R2(d) compares the ARPES second derivative plot and peak positions obtained from MDCs which are used for evaluating the c - f hybridization feature by PAM fitting. Since the MDC peaks almost perfectly follow the derivative image, we did not adopt the second-derivative method in our analysis.

Figure R2. ARPES second derivative plots of Figure 3(c) in the main text. The data were derived along (a) energy- and (b) momentum-directions. (c) MDCs near the E_F of Fig. 3(c) in the main text (d) the comparison between ARPES second derivative plot and peak positions obtained from MDCs in (c).

- The value of the effective mass obtained at low temperatures ($m^=6.56 \times 10^{-31}$) is smaller than the free electron mass. This makes the use of the term "heavy fermion" difficult to understand and should be explained by the authors. They claim to obtain a mass enhancement factor of about 120. They compare ARPES data at 300 K shown in Fig. S6 and those at 15 K in Fig. 3d, but I can't see much of a difference, particularly in the lower branch S2. Is the effective mass at low temperatures obtained from the upper branch S1? I think the authors should better explain how they have obtained the mass values for the bare and hybridized bands and include these values in the main text.*

Reply: We agree that the term "heavy fermion" in the current manuscript causes misreading because the obtained "heavy" effective mass is not heavier than the rest electron mass. We

are using this term that the electron mass at low temperature becomes heavier compared to high temperature due to the *c-f* hybridization. We moved an explanation of the evaluation of the effective mass in the main text from the supplemental note with an additional explanation.

- I think it would be helpful if the authors included in the supplementary information the ARPES intensity plots used to obtain Fig. 4. This will allow the reader to better assess the evolution of the band hybridization across the coherent temperature.

Reply: We agree with the comment. We add the ARPES intensity maps (Fig. R3 and we added the same figure in Fig. S5) used to obtain Fig. 4 to Supplemental Note 3.

Figure R3. Temperature-dependent ARPES intensity plots measured with horizontally polarized 35-eV photons. ARPES intensities are divided by the Fermi-Dirac distribution function convolved with the instrumental resolution.

- The authors state that “Perfect two-dimensional (2D) HF materials have not been reported yet”. Although they cited some previous work reporting the fabrication of artificial low dimensional strongly correlated electron systems, other topical references are lacking. For example, recent work published in Nature [Nature 616, 465-469 (2023)] reported lateral quantization of electronic states on the surface of a heavy fermion showing 2D heavy fermion behavior. I think this work should be mentioned to improve the introduction to this topic.

Reply: We appreciate the Reviewer’s suggestion about the introduction part. We added

some recently published references about 2D heavy fermion systems. We revised the sentence in the abstract and main text.

(abstract): However, **the realization of perfect two-dimensional (2D) HF materials is still a challenging topic.**

(main text): **Fabricating** artificial low-dimensional strongly correlated electron systems **and quantizing a three-dimensional HF state by quantum confinement [13]** are suitable methods to investigate the novel electronic phase.

Minor change:

- The sentence “The fitting results are shown in Fig. 2(c)” should be changed to “The fitting results are shown in Fig. 3(c)”.

Reply: Thank you for the careful review. We revised the sentence in the main text.

Reviewer #3 (Remarks to the Author):

The authors fabricate a 2D YbCu₂ heavy-fermion lattice on a clean Cu[111] surface by sputtering Yb atoms. LEED measurements confirm the high quality of the surface lattice. The authors then present a comprehensive ARPES study of this system which confirms the heavy-fermion behavior of this system. In particular, they demonstrate the k-dependent dispersion of the heavy band, the temperature dependent change of the Fermi momentum near the Gamma point, and the two-dimensionality of the heavy (flat) part of the band, while the conduction band must obviously be three-dimensionally due to the 3D Cu metal substrate. The measurements are of high quality, and the present system is one of the few truly two-dimensional heavy-fermion systems realized so far.

Therefore, I would in principle recommend publication of this manuscript, however would like to ask for clarification of some points.

Reply: First of all, let us express our deepest acknowledgment of the voluntary efforts of Prof. Kroha for our manuscript. We are encouraged that the reviewer evaluates our work as “The measurements are of high quality, and the present system is one of the few truly two-dimensional heavy-fermion systems realized so far. Therefore, I would in principle recommend publication of this manuscript.” We answer the questions, as shown in the following part.

(1) Apart from finding the absence of HF dispersion in the z direction (perpendicular to the surface), do the authors have independent evidence that the YbCu₂ lattice is truly a monolayer, as laimed?

Reply: Thanks for the comment. We evaporated excess Yb atoms onto Cu(111) to try to make a multilayer YbCu₂ and angle-integrated valence band spectra were measured, as shown in Figure R4. At binding energies of 0.2 eV and 1.4 eV, as indicated by arrows, additional peaks appeared overlaying on the spectrum of monolayer YbCu₂. These originated from doublet pairs of excess Yb atoms on YbCu₂ because pure Yb-metal shows a similar spectrum. So, we concluded that the surface alloying of YbCu₂ is limited to monolayer thickness. Further studies on real space imaging, such as STM, give us more information on the growth morphology of the YbCu₂ layer.

Figure R4. Angle-integrated photoelectron spectra of monoatomic layered YbCu₂ (black) and additionally Yb deposited monoatomic layered YbCu₂ (red).

*(2) Yb is a hole system (13 electrons, i.e., 1 hole, in the 4f shell in the pristine form). The hole-like nature is confirmed by the fact that the (k-integrated) ARPES spectra show the Kondo resonance below the Fermi energy, not above as for particle-like systems, which makes it clearly observable by ARPES, as seen in Fig. 4. Furthermore, the authors observe that the Fermi surface *expands* with lowering the temperature below the (Kondo) lattice-coherence temperature (Fig. 3b). Both observations together imply that also the conduction band is hole-like (this follows from counting mobile charge carriers or from an analysis where hybridized band must cross the Fermi level) and the fitting of the hybridized two-band structure (Eq. (1)) indeed yields a hole-like conduction band. By contrast, Cu is known to have particle-like conduction bands. Do they authors have an explanation for this apparent discrepancy? As this is an obviously emerging question, can they somewhat elaborate on it in the paper?*

Reply: Thanks for the valuable comments on the origin of the conduction band derived from Cu. As the reviewer said, generally, the electronic state of the nominal Cu shows an electron-like *sp*-band against the gamma point. On the other hand, the atom adsorbed (111) plane of the noble metal substrates such as Au, Ag, and Cu typically shows a $\sqrt{3}\times\sqrt{3}$ R30° periodicity and seems similar to hole-like bands at the gamma point. One hole band is derived from hybridization between adsorbed atoms and Cu. The others originate from the replica bands of bulk *sp*-band and the Umklapp scattering of *sp*-band from the $\sqrt{3}\times\sqrt{3}$ periodicity. Our DFT calculation also suggests that the inner hole band in this study mainly consists of the YbCu₂ layer (Fig. S2(b2)), and it shows the mixture of not only Yb 5*d*-orbitals but also Cu *sp*- and *d*-orbitals. This situation resembles other noble-metal $\sqrt{3}\times\sqrt{3}$ R30° systems such as other rare-

earth adsorbed ferromagnetic surface EuAu₂ [M. Blanco-Rey, PRR (2022).] and nodal line semimetal Cu₂Si/Cu(111) [B. Feng, NCommun. (2017).]. We added the explanation about the origin of the hole band to the supplemental note:

(Supplemental note6): The electronic state of the nominal Cu metal has an electron-like *sp*-band against the Γ point. On the other hand, atoms adsorbed on the (111) plane of noble metal substrates such as Au, Ag, and Cu typically form a $\sqrt{3}\times\sqrt{3}$ R30° periodicity and seem similar hole-like bands at the Γ point. The hole-like band is derived from hybridization between adsorbed atoms and Cu. The others originate from the replica bands of bulk *sp*-band and the Umklapp scattering of *sp*-band from $\sqrt{3}\times\sqrt{3}$ periodicity. Our DFT calculation also suggests that the inner hole band in this study mainly consists of the YbCu₂ layer, and it shows the mixture of orbitals not only Yb 5*d*-orbitals but also Cu *sp*- and *d*-orbitals, which is a similar situation for other noble-metal $\sqrt{3}\times\sqrt{3}$ R30° systems.

(3) Fig. 4 a, b shows a strong (almost temperature-independent below 30 K) decrease of spectral weight towards the Fermi energy, even after the spectra are divided by the Fermi-Dirac distribution. This seems to be a clear indication of a hybridization gap or even a complete gap, since this decrease occurs even in the k-integrated spectra. This should be discussed more explicitly in the paper.

Reply: Thanks for the valuable suggestion to explain our experimental results. We agree with the gap-opening scenario at E_F . We add the explanation about the hybridization gap based on the reviewer's comment in the main text as follows:

(main text): The spectral weight is strongly suppressed towards E_F , even after the spectra are divided by the Fermi-Dirac distribution function, suggesting that the *c-f* hybridization gap appears below $T_{coh.}$.

(4) The authors claim that gating would be an additional tuning parameter for 2D heavy-fermion systems. For the present system this would be difficult since the substrate is metallic. The authors should weaken their claim at least for the present system.

Reply: Thanks for the comment. Of course, we understand that metallic substrate is not suitable for gating. The fabrication of single crystalline Cu thin films on insulating materials such as sapphire and Si(111) may be one answer to this concern. We revised the manuscript as follows:

(main text): The ground state of YbCu₂ would be tuned around QCP by external perturbations conventionally applied to other 2D materials, where surface carrier doping by alkali metal adsorption and gate-tuning of carrier concentration by the growth of Cu(111) ultrathin film on an insulating substrate such as sapphire may be realized.

Such novel techniques are expected to explore novel quantum critical phenomena in 2D materials, such as atomic-layer unconventional superconductivity.

(5) The authors find an increase of the Kondo spectral weight even at temperatures below the coherence (or Kondo) temperature and quote this as unusual. In fact however, the coherence temperature T_{coh} sets the temperature scale on which the weight increases, and saturation typically occurs below T_{coh} . This is particularly visible in spectral properties like ARPES, while response quantities, like conductivity, specific heat etc, are less clear. It has been shown by Bethe ansatz solution of the single-impurity problem and numerous calculations for lattice problems.

Johann Kroha

Reply: We thank Prof. Kroha for providing a better explanation of the temperature-dependence Kondo resonance peak by ARPES. We completely agree with the explanation then it adds the sentences of comments in the main text:

(main text) T_{coh} sets the temperature scale on which the weight increases and saturation typically occurs below T_{coh} . This result is evidently visible in spectral properties like ARPES, while response quantities, such as conductivity and specific heat, are less clear. It has been shown by Bethe ansatz solution of the single-impurity problem and numerous calculations for lattice problems.

Reviewers' Comments:

Reviewer #1:

Remarks to the Author:

The authors have taken into account the different comments/notes raised by all reviewers. It is my opinion that the present and revised version of their manuscript can be now accepted for publication.

Further, I think the reported results on a monoatomic layer of YbCu₂ are new and reliable.

Reviewer #2:

Remarks to the Author:

After reading the resubmitted version of the paper and authors' response, I believe that my main concerns have been convincingly answered. The experimental data are of very good quality and the fabrication and study of a 2D heavy fermion system is original and supported by the experiments. I believe that this work will be of interest to the broad audience of Nature Communications and therefore recommend its publication.

Reviewer #3:

Remarks to the Author:

The authors have convincingly responded to my questions and have made appropriate changes to the manuscript, in particular regarding the point about the hole-like dispersion of the host system. Therefore, I can now recommend the paper in its present form.

Johann Kroha

Reviewer #1 (Remarks to the Author):

The authors have taken into account the different comments/notes raised by all reviewers. It is my opinion that the present and revised version of their manuscript can be now accepted for publication.

Further, I think the reported results on a monoatomic layer of YbCu₂ are new and reliable.

Reply: We are grateful to Reviewer #1 for the strong recommendation for the publication of this manuscript in Nat. Commun. We sincerely thank the reviewer for his/her valuable suggestions to improve our manuscript through the review.

Reviewer #2 (Remarks to the Author):

After reading the resubmitted version of the paper and authors' response, I believe that my main concerns have been convincingly answered. The experimental data are of very good quality and the fabrication and study of a 2D heavy fermion system is original and supported by the experiments. I believe that this work will be of interest to the broad audience of Nature Communications and therefore recommend its publication.

Reply: We thank Reviewer #2 for supporting our manuscript to publish in Nature Communications. We sincerely thank the reviewer for his/her valuable suggestions to improve our manuscript through the review.

Reviewer #3 (Remarks to the Author):

The authors have convincingly responded to my questions and have made appropriate changes to the manuscript, in particular regarding the point about the hole-like dispersion of the host system. Therefore, I can now recommend the paper in its present form.

Johann Kroha

Reply: Let us express our deepest acknowledgement to Prof. Kroha for publishing in Nature Communications. we could improve our manuscript significantly.